# Heterotrophic Cultivation of *Euglena gracilis* in Stirred Tank Bioreactor: A Promising Bioprocess for Sustainable Paramylon Production

**DOI:** 10.3390/molecules27185866

**Published:** 2022-09-09

**Authors:** Franjo Ivušić, Tonči Rezić, Božidar Šantek

**Affiliations:** 1Croatian Academy of Sciences and Arts, Vlaha Bukovca 14, 20000 Dubrovnik, Croatia; 2Faculty of Food Technology and Biotechnology, University of Zagreb, Pierottijeva 6, 10000 Zagreb, Croatia

**Keywords:** *Euglena gracilis*, paramylon, stirred tank bioreactor, heterotrophic cultivation, corn steep solid

## Abstract

Paramylon is a valuable intracellular product of the microalgae *Euglena gracilis*, and it can accumulate in *Euglena* cells according to the cultivation conditions. For the sustainable production of paramylon and appropriate cell growth, different bioreactor processes and industrial byproducts can be considered as substrates. In this study, a complex medium with corn steep solid (CSS) was used, and various bioreactor processes (batch, fed batch, semicontinuous and continuous) were performed in order to maximize paramylon production in the microalgae *Euglena gracilis*. Compared to the batch, fed batch and repeated batch bioprocesses, during the continuous bioprocess in a stirred tank bioreactor (STR) with a complex medium containing 20 g/L of glucose and 25 g/L of CSS, *E. gracilis* accumulated a competitive paramylon content (67.0%), and the highest paramylon productivity of 0.189 g/Lh was observed. This demonstrated that the application of a continuous bioprocess, with corn steep solid as an industrial byproduct, can be a successful strategy for efficient and economical paramylon production.

## 1. Introduction

Microalgae contain several high-value molecules, such as lipids, proteins and carbohydrates. These products can efficiently accumulate in microalgae biomass during the cultivation process. For this reason, there is a growing interest in microalgae as production organisms and the development of an effective cultivation bioprocess [1,2,3,4,5].

*Euglena gracilis* is a microalga with the potential to accumulate large amounts of the reserve polysaccharide β-1,3-glucan, known as paramylon [6]. Paramylon is a discoidal granule, similar to starch, that has high crystallinity (~90%), and its content often exceeds 50% of the dry weight of the cell, especially under heterotrophic growth conditions [7] (Tanaka et al., 2017). The higher quality of *E. gracilis* β-glucan (paramylon) molecules compared to other sources of glucan is mainly due to a chemical conformation that is much more efficient in the promotion of immunological functions [8]. As a value-added product, paramylon is utilised as an important product for food additives and pharmaceutical industries. *Euglena gracilis* has been cultivated heterotrophically and photoautotrophically on various media compositions, ranging from simple and chemically defined formulae to complex media with industrial byproducts: molasses, corn steep solids and yeast extract [9]. The major challenges are in finding low-cost media with appropriate substrates. Reports from previous research also indicate an effective utilisation of byproducts from food and beverage processing plants as a substrate for *E. gracilis* cultivation. Examples of the efficient utilisation of byproducts are: potato liquor [10], corn steep solid [11], ferulic acid from rice bran [12] and spent tomato byproduct [1]. As an industrial byproduct generated during corn wet-milling, corn steep solid (CSS) has been successfully utilised in microbial biomass production. Its potential in the preparation of complex substrates for microbial production has been reported for: lactic acid production by *Lactobacillus rhamnosus*, cellulolytic enzyme production by *Streptomyces malaysiensis* and paramylon production in *Euglena* species [11,13,14]. CSS is an excellent source of nitrogen for microbial growth, as well as a source of organic acids such as lactic acid, polypeptides, amino acids and B-complex vitamins [13].

To the best of our knowledge, studies regarding industrial byproduct utilisation and different bioprocess applications for paramylon production are highly limited. The production of *E. gracilis* is strongly dependent not only on the substrate types, but also on the cultivation conditions, such as trophic mode, pH, temperature and nutritional content. Additionally, the bioreactor type and bioprocess mode selection have a strong influence on *E. gracilis* biomass production and intracellular product accumulation [15,16].

Regarding the various bioprocess modes (e.g., batch, fed batch, repeated batch and continuous), suitable medium compositions can differ substantially, and the efficiency of bioprocesses can be improved through the selection of appropriate media and bioprocess modes [17,18]. In the continuous bioprocess mode, flow rates of a fresh medium (inflow rate) and used medium with microbial cells (outflow rate) have to be harmonised. To reach steady-state flow rates (dilution rates), processes have to be adapted according to the microbial cells growth rates. In the steady state, cultivation conditions remain constant over extended periods of time, thus, upstream costs are greatly reduced and processes are more economically suitable [19]. Based on a constant power to volume ratio, a continuous bioprocess can be easy scaled-up [20].

The objective of this study was to investigate the applicability of industrial byproducts, corn steep solids (CSS) and various bioreactor process modes on the heterotrophic cultivation of *E. gracilis* and paramylon production in a stirred tank bioreactor. Batch, fed batch, repeated batch and continuous bioprocess modes were evaluated with various bioprocess efficiency parameters (concentrations, yields and productivities of *E. gracilis* biomass and paramylon production). The most appropriate bioprocess mode for paramylon production was selected.

## 2. Results and Discussion

After the formulation of a suitable growth medium in a laboratory shaker [11], it was necessary to confirm the obtained results on a larger scale; therefore, cultivation in a laboratory stirred tank bioreactor was performed. In this study, cell losses were at negligible levels. Furthermore, water evaporation losses were also noticed, and, consequently, every 24 h in the STR, approximately 250 mL of sterile water was added. Two steps were conducted; in the first step, batch experiments with different growth media were tested, and heterotrophic cultivations of *E. gracilis* in a stirred tank bioreactor were performed. In the second step, different bioprocess modes (fed batch, repeated batch and continuous) were tested with the selected medium.

### 2.1. Selection of Complex Media Compositions for E. gracilis Cultivation in a Stirred Tank Bioreactor

To select the most appropriate complex medium for the heterotrophic cultivation of *E. gracilis* in a stirred tank bioreactor (STR), five batch bioprocesses were performed by using complex media containing 25 g/L of CSS and 20 g/L of glucose, fructose, galactose or sucrose. For comparison, one batch was performed with the Hutner medium (Figure 1a).

Batch cultivations in a stirred tank bioreactor, with the addition of 20 g/L glucose as a carbon source and 25 g/L CSS addition as a nitrogen source, are presented in Figure 1b. *E. gracilis* biomass growth and paramylon synthesis were directly connected to the consumption of glucose and nitrogen compounds from the CSS (Figure 1b). When compared to the experiment on Hutner medium (Figure 1a), substantially higher biomass and paramylon concentrations were recorded, which indicated that CSS was a suitable source of nitrogen compounds and growth factors for *E. gracilis* cultivation. Moreover, the highest values were recorded after 90 h (*X_M_* = 15.8 g/L; *P_M_* = 8.5 g/L) of cultivation (on Hutner medium, it was after 136 h; *X_M_* = 12.4 g/L; *P_M_* = 5.7 g/L). Changes in the soluble dry weight (DW) during the bioprocess were in accordance with the changes in glucose (glucose source) concentrations. Similar phenomena were observed throughout the whole research. However, the *E. gracilis* biomass concentration was slightly higher compared to previous cultivations with potato liquor, but the paramylon concentration was lower [21,22,23].

The heterotrophic cultivation of *E. gracilis* in a stirred tank bioreactor was also performed on a growth medium with 20 g/L fructose and 25 g/L CSS. The results of this cultivation with the addition of fructose are presented in Figure 1c. Compared to the experiment with the glucose addition, in this cultivation, lower biomass and paramylon yields were recorded (*X_M_* = 12.1 g/L; *P_M_* = 8.6 g/L). It could be concluded that glucose was a more adequate carbon source for *E. gracilis* growth and paramylon synthesis. Moreover, in this experiment, productivities were lower because maximum concentrations were recorded after 120 h of cultivation (in glucose experiment after 90 h).

The heterotrophic cultivation of *E. gracilis* was further performed on a medium with 20 g/L galactose and 25 g/L CSS. The results of this cultivation are presented in Figure 1d. It was clear that the *E. gracilis* biomass did not utilise galactose for biomass growth and paramylon synthesis. The moderate increase in biomass and paramylon concentration was due to the consumption of CSS components. Therefore, concentrations recorded in this cultivation (*X_M_* = 5.5 g/L; *P_M_* = 2.0 g/L) were significantly lower when compared to cultivations with a glucose or fructose addition. Concentrations recorded in this cultivation were similar to concentrations achieved in shake flasks, but also significantly lower when compared to cultivations with a glucose or fructose addition [11].

Finally, the heterotrophic cultivation of *E. gracilis* was performed in a STR on a medium with 20 g/L sucrose and 25 g/L CSS. The results of this cultivation are presented in Figure 1e. It was clear that the *E. gracilis* biomass did not utilise disaccharide sucrose for biomass growth and paramylon synthesis. The moderate increase in biomass and paramylon concentration (*X_M_* = 4.8 g/L; *P_M_* = 2.5 g/L) was similar to a previous experiment on galactose.

CSS was used as a source of complex nitrogen, vitamins and other growth factors, as well as a suitable nitrogen source for the heterotrophic cultivation of *E. gracilis.* Many authors emphasize that vitamins, especially B1 and B12, are important factors for a successful scale-up and contribute to the overall paramylon production costs [6,24,25,26]. Therefore, the efficient supplementation of vitamins from a cheap source, such as CSS, is an important step in the development of paramylon production systems.

Results from Figure 1 confirmed glucose and fructose as the most suitable carbon source for the successful cultivation of *E. gracilis* and paramylon production in a STR. However, compared to fructose, glucose was selected as the carbon source for further investigations, due to its lower world market price [11]. It was observed that galactose and sucrose were metabolized very slowly, and, therefore, biomass and paramylon concentrations were relatively low compared to the batch cultivation with glucose and fructose (Figure 1). Prior to cultivation in a STR, sucrose can be pretreated and hydrolysed to produce fructose and glucose carbon sources. The solution could also be a genetic modification of *E. gracilis* to promote the synthesis of enzymes for sucrose transfer and hydrolysis in *E. gracilis* cells. This solution had obstacles regarding the regulation and utilisation of genetically modified organisms [6].

Bioprocess conditions used for *E gracilis* cultivation in the STR on Hutner (Batch a) and complex media containing 25 g/L CSS and 20 g/L of glucose (Batch b), fructose (Batch c), galactose (Batch d) and sucrose (Batch e) are summarised in Table 1. Table 1 also contains bioprocess efficiency parameters calculated from the data presented in Figure 1. Bioprocess efficiency parameters were used for the selection of the most appropriate medium for paramylon production in the STR. The highest productivity for the *E. gracilis* biomass and paramylon production was calculated for the batch conducted with the complex medium containing 25 g/L of CSS and 20 g/L of glucose. This medium was used in the next set of experiments to select the most appropriate bioprocess mode and enhance the bioprocess efficiency.

### 2.2. Selection of Bioprocess Mode for E. gracilis Cultivation in Stirred Tank Bioreactor

The utilisation of the medium’s nutrients during batch cultivation affects the growth of *E. gracilis* and paramylon production. To prevent the effects of nutrient medium limitations and increase biomass and paramylon production, different bioprocess modes were tested. A continuous process, as well as the fed batch and repeated batch processes, can be applied to regulate substrate concentrations and enhance process productivity [6].

To select the most appropriate bioprocess mode and enhance the productivity of paramylon production, the heterotrophic cultivation of *E. gracilis* was performed in a stirred tank bioreactor in 20 g/L glucose and 25 g/L CSS. The results of *E. gracilis* cultivation in a stirred tank bioreactor using various bioprocess modes (fed batch, repeated batch and continuous) are presented in Figure 2.

The results of fed batch cultivation are shown in Figure 2a. The highest biomass (approximately 20 g/L) and paramylon (11–17.5 g/L) concentrations were obtained between 125 and 154 h of cultivation. Those values were higher than those obtained during batch cultivation on the same medium (Figure 1b). Glucose concentration was maintained within a range of 3–5 g/L with the appropriate feeding rate and glucose consumption. Heterotrophic fed batch cultivation started in the STR in 20 g/L glucose and 25 g/L CSS. For feeding, the medium was concentrated three times, and the fed batch started after approximately 96 h of batch cultivation, which coincided with the onset of the stationary growth phase. In this case, the initial medium dry weight (DW) was lower (approx. 22 g/L) than in all other bioprocess modes. A possible explanation may be CSS heterogeneity as a medium constituent. Furthermore, it has to be pointed out that CSS and all other complex medium constituents were always homogenised before they were used for medium preparation. Bioreactor sterilization was performed with indirect steam, and, therefore, the steam condensation effect (dilution effect) in the medium could be neglected. Feeding rates depended on the medium nutrient consumption rates, and it was employed in order to maximise yields of paramylon at elevated biomass concentrations.

Using the same complex medium with 20 g/L glucose and 25 g/L CSS, a repeated batch (semicontinuous) cultivation of *E. gracilis* was performed. After initial batch cultivation (approximately 96 h of cultivation), 6.7 L (67%) of spent medium was replaced with fresh medium of the same composition. The replacement of the medium was performed at the moment when glucose concentration diminished below 5 g/L. Five medium replacements were performed in order to achieve a stable bioprocess performance. The results of the repeated batch cultivation are given in Figure 2b. When compared to fed batch cultivation, the acquired biomass and paramylon concentrations were negligibly lower, and glucose uptake by *E. gracilis* was complete. These findings showed that a complex medium with 20 g/L glucose and 25 g/L CSS was suitable for the repeated batch cultivation of *E. gracilis* biomass.

In the next experiment, a continuous cultivation of *E. gracilis* was performed on a complex medium with 20 g/L glucose and 25 g/L CSS. Since the maximum specific growth rate of *E. gracilis* was approximately 0.05 1/h [27], for this purpose, the dilution rate was set to 0.04 1/h. After the batch cultivation (approximately 96 h of cultivation), the continuous mode started with an inflow rate of 0.2 L/h, which corresponded to a 0.04 1/h dilution rate. However, after a short period, it was evident that concentrations of biomass and paramylon had a tendency to decrease, i.e., the dilution rate was too high, so the inflow rate was decreased to 0.135 L/h (D = 0.027 1/h) in order to maintain a glucose concentration in the effluent of approximately 5 g/L. Once the steady state was established, concentrations of biomass and paramylon in the effluent were lower for the fed batch and repeated batch cultivations.

In order to summarize the results, a comparison of different cultivation modes was performed. Results that were achieved on complex medium with 20 g/L glucose and 25 g/L CSS were compared separately in terms of bioprocess mode efficiency (Table 2). Maximum *E. gracilis* biomass (*X*_M_ = 19.4 g/L) and paramylon (*P*_M_ = 17.5 g/L) concentrations showed that the most favourable bioprocess mode regarding the highest *E. gracilis* biomass concentration and paramylon accumulation was the fed batch process. As for the paramylon and biomass productivity, the most favourable bioprocess mode was the continuous process, with biomass productivity (*Pr_X_*) of 0.284 g/Lh and paramylon productivity (*Pr_P_*) of 0.189 g/Lh. Productivities were in the range of previously optimised fed batch bioprocesses for paramylon production in a STR on complex medium with potato liquor [10]. Obstacles in the continuous bioprocess mode were the lower biomass concentration (*X*_M_ = 10.67 g/L) and relatively high concentration of glucose (G) and soluble dry weight (DW) content in the outflow (Figure 2c). The increase in medium components in the outflow and relatively low biomass concentration had negative impacts on the downstream processes and paramylon purification. However, a continuous cultivation can be easy prolonged, as well as the nature of the continuous operation reduced in time and costs for the upstream processes [19,20].

## 3. Materials and Methods

### 3.1. Microorganisms and Cultivation Media

*Euglena gracilis* strain Z (Klebs SAG 1224-5/25) was obtained from the Algensammlung Göttingen, Germany. *E. gracilis* was maintained on modified Hutner medium [11]. In this research, the Hutner and complex media (consisting of 20 g/L of glucose (Kemika, Croatia), fructose (Kemika, Croatia), galactose (Kemika, Croatia) and sucrose (Kemika, Croatia) and 25 g/L of corn steep solid (CSS) (Roquette, French)) were used for the heterotrophic cultivation of *E. gracilis*. All media were sterilized at 121 °C for 20 min, then cooled down prior to inoculation. Inocula of *E. gracilis* were prepared separately on the medium (Hutner or complex medium) that was used for further research. Inocula were propagated on a rotary shaker (DIY, Croatia) in 500 mL Erlenmeyer flasks with 300 mL medium at 28 °C for 72 h and a rotation speed of 150 min^−1^. The cell number concentration in the *E. gracilis* inoculum was approx. 10^7^ CFU/mL.

### 3.2. Heterotrophic Cultivation of E. gracilis in the STR under Different Bioprocess Modes

All experiments were performed after sterilization (together, medium and bioreactor) at 121 °C for 20 min and subsequent cooling to the working temperature. All cultivations were performed in a stirred tank bioreactor (Sartorius Biostat Cplus, Göttingen, Germany) at 28 °C with a rotation speed of 250 rpm and aeration rate in the range of 0.3–0.8 v/vmin in order to achieve a 30–40% air saturation of the medium. The working volume for the batch was 10 L. For *E. gracilis* cultivation on the Hutner and complex media, the 10% (*v*/*v*) of inocula that contained the cell suspension of *E. gracilis* was prepared. To ensure heterotrophic growth, the bioreactor window was wrapped in aluminium foil in order to avoid light penetration. Fed batch, repeated batch and continuous bioprocess cultivation modes started in a batch mode, and nutrient feeding and broth harvest were carried out with a set of pumps.

Fed batch cultivation started in 7 L of working volume and the same conditions as for the batch cultivation. After 4 days of batch cultivation, feeding started with the addition of fresh medium concentrated three times. In the fed batch cultivation mode, fresh medium was added when the glucose concentration decreased to under 5 g/L.

Repeated batch cultivation started in 10 L of working volume and the same conditions as for the batch cultivation. After 4 days of batch cultivation, a repeated batch was performed by the replacement of 6.7 L of spent medium with the same volume of fresh medium at the moment when the concentration of glucose fell under 5 g/L. The fresh medium had the same composition and concentration as the starting medium. The replacement of spent medium with a fresh one was performed five times consecutively during repeated batch cultivation.

Continuous cultivation started in 5 L of working volume and the same conditions as for the batch cultivation. After 4 days of batch cultivation, continuous cultivation was performed. To set the investigated dilution rate (0.027 1/h) for the continuous cultivation mode, the feeding of fresh medium and harvesting of spent medium started at the same time with a set of two pumps (one for feeding and the other one for harvesting). During continuous cultivation, the working volume was changed five times, and cultivation was performed for 15 days [28].

The monitoring of bioprocess performance was performed by withdrawing the broth samples during all *E. gracilis* cultivations.

### 3.3. Analytical Procedures and Bioprocess Efficiency

The determination of biomass concentrations was performed gravimetrically. For this purpose, homogenized samples were centrifuged at 3629× *g* (MSE, Harrier 18/80, London, UK) for 15 min and pellets were dried at 75 °C to a constant mass. Supernatants were used for the determination of sugar concentrations (S) and total soluble dry weight (DW). Glucose (G) concentration was determined with a glucose enzymatic assay kit, supplied by Sigma-Aldrich, St. Louis, MO, USA. Other sugar concentrations were measured with the Anthrone reagent method [18]. The total soluble dry weight was measured with drying supernatant aliquots (5 mL) at 60 °C to a constant weight. Paramylon (P) was extracted from biomass suspensions (10 mL) with ultrasonic disintegration (1–2 min; 20 kHz; max 0.2 kW, Bandelin electronic, Berlin, Germany). The ultrasound disintegration of *E. gracilis* cells were checked after each treatment (1 min) under a microscope and repeated if intact cells were found in the treated sample. After a second ultrasound treatment, for all samples, intact algae cells were not found. The purification of disintegrated biomass was conducted by resuspending cells in a solution containing 1% (*m*/*V*) sodium dodecyl sulphate (SDS) (Kemika, Zagreb, Croatia). This suspension was incubated for 2 days at 37 °C, and the paramylon granules were recovered with centrifugation for 15 min at 4000 rpm. Centrifugation was repeated as paramylon was washed twice with demineralized water. After the second wash, the granules were dried at 60 °C to a constant weight.

Bioprocess efficiency parameters (paramylon conversion coefficient (*Y_P/X_*) and productivity (*Pr*)) were calculated with standard procedures [11].

The paramylon conversion coefficient (*Y_P/X_*) was estimated with the following equation:*Y_P_*_/_*_X_* = *P* − *P*_0_/(*X* − *X*_0_)(1)
where *P* and *P*_0_ are paramylon concentrations at the end and at the beginning of the bioprocess; *X_0_* and *X* are *E. gracilis* biomass concentrations at the beginning and at the end of the bioprocess, respectively.

Bioprocess productivities for biomass and paramylon production (*Pr_X_* and *Pr_P_*) were determined with the following equations for the batch, fed batch and repeated batch cultivations:*Pr_X_* = *X* − *X*_0_/*t*(2)
*Pr_P_* = *P* − *P*_0_/*t*(3)
where *t* is cultivation time.

For continuous cultivation, productivity was calculated using the following equation:*Pr_X_* = *X*·*D*(4)
*Pr_P_* = *P*·*D*(5)
where *X* is the biomass concentration (or paramylon concentration, *P*) in the outflow and *D* is the dilution rate calculated with the inflow rate and the bioreactor’s working volume (10 L of suspended microalgae).

## 4. Conclusions

Glucose was the most efficient carbon source for the successful cultivation of *E. gracilis* and paramylon production in a STR. Galactose and sucrose were very slowly metabolized by *E. gracilis,* and, therefore, these carbon sources were not suitable for paramylon production on a large scale. Corn steep solid (CSS) was a suitable source of nitrogen and growth factors for the heterotrophic cultivation of *E. gracilis.*

A batch cultivation of *E. gracilis* and paramylon production could be successfully conducted in a STR on complex medium consisting of 20 g/L glucose and 25 g/L CSS as a substitute for chemically defined Hutner medium. According to the efficiency parameter calculations, the most efficient bioprocess modes for *E. gracilis* cultivation and paramylon production were the fed batch and continuous modes. On the basis of the productivity calculation results, it was obvious that the continuous bioprocess mode has great potential for the industrial production of paramylon by *E. gracilis*.

## Figures and Tables

**Figure 1 molecules-27-05866-f001:**
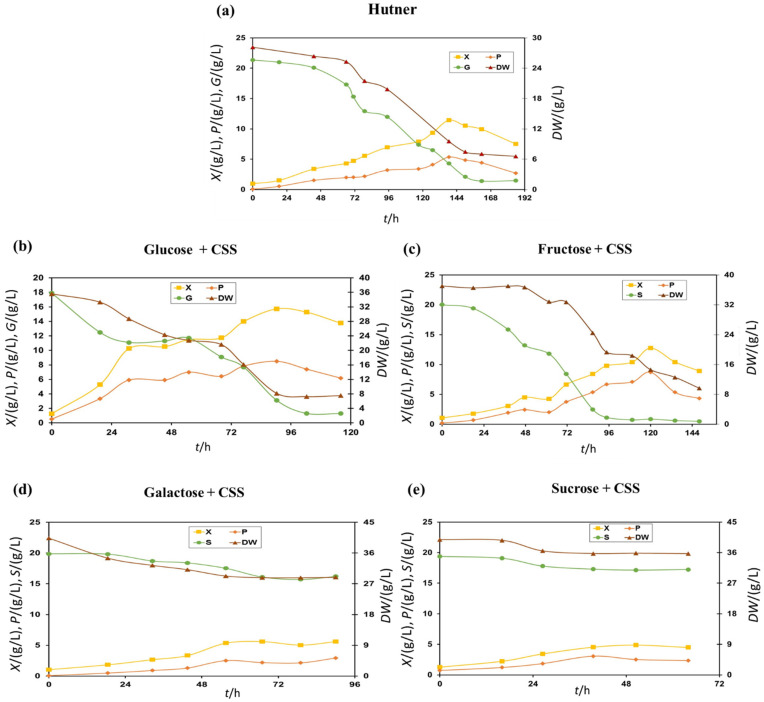
Batch cultivation of *E. gracilis* in a stirred tank bioreactor on Hutner medium and complex medium with different carbon sources: (**a**) Hutner medium; (**b**) medium with glucose and CSS; (**c**) medium with fructose and CSS; (**d**) medium with galactose and CSS; (**e**) medium with sucrose and CSS.

**Figure 2 molecules-27-05866-f002:**
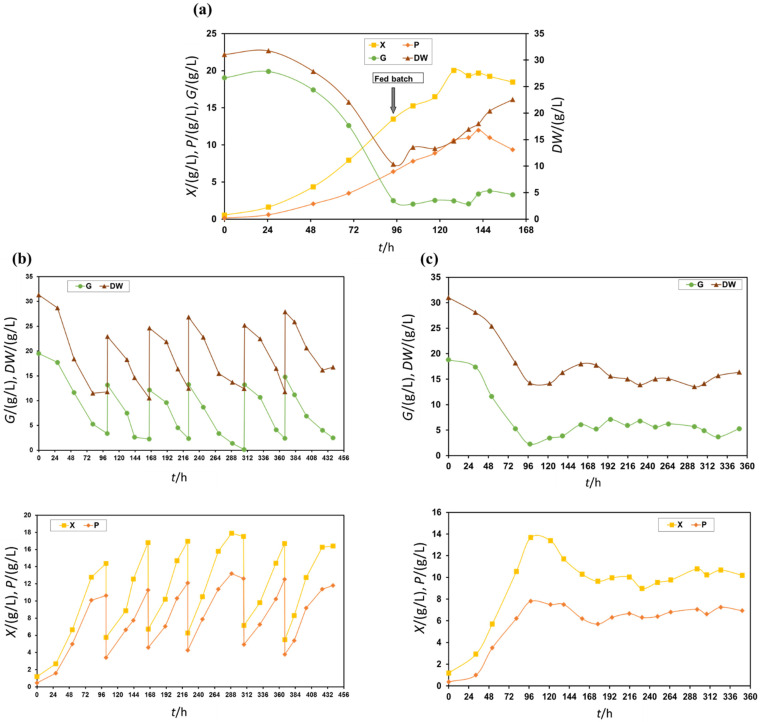
Heterotrophic cultivation of *E. gracilis* and paramylon production in a stirred tank bioreactor with different bioprocess modes: (**a**) fed batch; (**b**) repeated batch; (**c**) continuous.

**Table 1 molecules-27-05866-t001:** Bioprocess efficiency parameters used for *E. gracilis* batch cultivation in a STR on chemically defined Hutner medium (batch a) and complex media containing 25 g/L of CSS and 20 g/L of: glucose (batch b), fructose (batch c), galactose (batch d) and sucrose (batch e).

Bioprocess Mode	*t_M_*/h	*X_M_*/g/L	*P_M_*/g/L	*Y_P/X_*/g/g	*Pr**_X_*/g/Lh	*Pr_P_*/g/Lh
	Hutner					
Batch a	136	12.4	5.7	0.46	0.091	0.042
	Glucose and CSS					
Batch b	90	15.8	8.5	0.54	0.176	0.094
	Fructose and CSS					
Batch c	120	12.1	8.6	0.71	0.100	0.071
	Galactose and CSS					
Batch d	56	5.5	2.0	0.36	0.098	0.036
	Sucrose and CSS					
Batch e	40	4.8	2.5	0.52	0.120	0.062

*t_M_* time when maximum paramylon (*P_M_*) concentration and corresponding biomass concentration (*X_M_*) were observed, *Y_P/X_* paramylon conversion coefficient, *Pr_X_* bioprocess productivity for biomass production, *Pr_P_* bioprocess productivity for paramylon production.

**Table 2 molecules-27-05866-t002:** Bioprocess efficiency parameters calculated during heterotrophic cultivation of *E. gracilis* with different bioprocess modes in a complex medium containing 25 g/L of CSS and 20 g/L of glucose.

Bioprocess Mode	*t_M_*/h	*X_M_*/g/L	*P_M_*/g/L	*Y_P/X_*/g/g	*Pr**_X_*/g/Lh	*Pr**_P_*/g/Lh
Fed batch	154	19.4	17.5	0.90	0.126	0.113
Repeated batch	446	17.9(Σ99.4)	13.2(Σ71.2)	0.72	0.222	0.160
Continuous	288	10.5	7.0	0.67	0.284	0.189

*t_M_* time when maximum paramylon (*P_M_*) concentration and corresponding biomass concentration (*X_M_*) were observed; for repeated batch *Pr**_X_* and *Pr**_P_*, a maximum concentration for all stages was counted, *Y_P/X_* paramylon conversion coefficient.

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
