# Peer review of "Heterotrophic Cultivation of Euglena gracilis in Stirred Tank Bioreactor: A Promising Bioprocess for Sustainable Paramylon Production"

_molecules, 2022, doi:10.3390/molecules27185866_

Round 1

Reviewer 1 Report

The manuscript entitled, “Heterotrophic Cultivation of Euglena gracilis in Stirred Tank Bioreactor: A Promising Bioprocess for Sustainable Paramylon Production” has studied to performed to various bioreactor processes to maximize paramylon production in microalgae Euglena gracilis in stirred tank bioreactor (STR) with containing glucose and corn steep solid (CSS) medium.  

Euglena gracilis can accumulate large amounts of the reserve polysaccharide paramylon, a β-1,3- glucan, which can constitute over 80% (w/w) of the dry weight (DW, biomass dried to a constant weight without oxidation). https://doi.org/10.3389/fbioe.2019.00108

Euglena gracilis has been cultivated heterotrophically and photoautotrophically on various nutrient compositions, ranging from simple and chemically defined formulae to complex media with industrial byproducts (e.g., molasses, corn steep solids, and yeast extract) to accumulate 50–75% of the cell biomass as paramylon. https://doi.org/10.1111/jpy.12758

PM is a discoidalgranule that has high crystallinity (~90%) and its con-tent often exceeds 50% of the dry weight of the cell, especially under heterotrophic growth conditions doi:10.1002/1873-3468.12659.

-This study was compared to other articles, and it achieved results that the continuous bioprocess in stirred tank bioreactor (STR) with complex medium, containing 20 g/L of glucose and 25 g/L of CSS, E. gracilis accumulate 67.0% paramylon content.

I think the article will gain a stronger scientific value after the suggested revisions. Correction regarding the following few comments:

“Complex medium is consisting of 20  g/L of glucose, fructose, galactose and sucrose and 25 g/L of corn steep solid (CSS) were used for heterotrophic cultivation of E. gracilis”. All monosaccharides and disaccharide should have been added to the complex medium, or prepared it separately. Because table 1 has seen that the complex medium containing 25 g/L of CSS and 20 g/L of: glucose, fructose, galactose and sucrose in prepared hunter medium or distilled water, respectively.

10 L of working volume was used for the repeated batch and continuous mode. (Inoculum concentration should have been used at only 300 mL for 10 L STR production) For E. gracilis cultivation on Hutner and complex medium the 300 mL inocula that contained the cell suspension of E. gracilis was prepared.

Result and Discussion:

 Nevertheless, it needs substantial English language editing and correction regarding following comments:

-bigger scale or large scale.

Table 1  Fructose and CSS XM/g/L and PM/g/L amounts are higher than Glucose and CSS at batch cultivation. Why you have chosen Glucose and CSS medium?

And then Figure 1 b. X,P, and G (g/L) concentration why should be initiated 18.

Figure 2. Heterotrophic cultivation of E. gracilis and paramylon production in stirred tank bioreactor by the different bioprocesses modes: a) fed batch (DW should start22 g/L whereas other processes b, and c start at 32 g/L dry weight ); b) repeated batch; c) continuous.

The discussion and conclusion should be strengthened.

Author Response

Dear Editor,

we would like to thanks reviewer for the insightful comments and suggestions which were very useful during revision process. In general, the whole manuscript was carefully read and corrected (major changes are in red color) according to the reviewer’s suggestions. New references are added and consequently according to the authors instructions. Our comments to reviewer suggestions are presented below.

Reviewer 1:

The manuscript entitled, “Heterotrophic Cultivation of Euglena gracilis in Stirred Tank Bioreactor: A Promising Bioprocess for Sustainable Paramylon Production” has studied to performed to various bioreactor processes to maximize paramylon production in microalgae Euglena gracilis in stirred tank bioreactor (STR) with containing glucose and corn steep solid (CSS) medium.  

Euglena gracilis can accumulate large amounts of the reserve polysaccharide paramylon, a β-1,3- glucan, which can constitute over 80% (w/w) of the dry weight (DW, biomass dried to a constant weight without oxidation). https://doi.org/10.3389/fbioe.2019.00108

Euglena gracilis has been cultivated heterotrophically and photoautotrophically on various nutrient compositions, ranging from simple and chemically defined formulae to complex media with industrial byproducts (e.g., molasses, corn steep solids, and yeast extract) to accumulate 50–75% of the cell biomass as paramylon. https://doi.org/10.1111/jpy.12758

PM is a discoidal granule that has high crystallinity (~90%) and its content often exceeds 50% of the dry weight of the cell, especially under heterotrophic growth conditions doi:10.1002/1873-3468.12659.

Our comment: Thank you for the suggested references. Revised version of the manuscript is improved by the indicated citations and the related comments (see lines 79-83; 87-41).

This study was compared to other articles, and it achieved results that the continuous bioprocess in stirred tank bioreactor (STR) with complex medium, containing 20 g/L of glucose and 25 g/L of CSS, E. gracilis accumulate 67.0% paramylon content.

I think the article will gain a stronger scientific value after the suggested revisions. Correction regarding the following few comments:

“Complex medium is consisting of 20 g/L of glucose, fructose, galactose and sucrose and 25 g/L of corn steep solid (CSS) were used for heterotrophic cultivation of E. gracilis”. All monosaccharides and disaccharide should have been added to the complex medium, or prepared it separately. Because table 1 has seen that the complex medium containing 25 g/L of CSS and 20 g/L of: glucose, fructose, galactose and sucrose in prepared hunter medium or distilled water, respectively.

Our comment: In the presented manuscript all monosaccharides and disaccharide have been added to the complex medium separately. Batches with different carbon source is now indicated in the Table 1 as: Batch a, b c d and e (see lines 241-244).

10 L of working volume was used for the repeated batch and continuous mode. (Inoculum concentration should have been used at only 300 mL for 10 L STR production) For E. gracilis cultivation on Hutner and complex medium the 300 mL inocula that contained the cell suspension of E. gracilis was prepared.

Our comment: We agree with this comment and consequently in revised manuscript our mistake was corrected. The correct quantity of inoculum in this study was 10 % v/v. (see lines 148-150).

Result and Discussion:

Nevertheless, it needs substantial English language editing and correction regarding following comments:

Our comment: We agree with this comment. The manuscript language was corrected by experience English speaker.

-bigger scale or large scale.

Our comment: We agree with this comment and consequently in the whole manuscript expression “bigger scale” was replaced by expression ”larger scale”.

Table 1 Fructose and CSS XM/g/L and PM/g/L amounts are higher than Glucose and CSS at batch cultivation. Why you have chosen Glucose and CSS medium?

Our comment: Thank you for the observation according the indicated XM and PM amounts. In the revised manuscript data are replaced and corrected according the cultivation results presented in the Figure 1. (see lines 179, 180, 189,198, 206)

And then Figure 1 b. X,P, and G (g/L) concentration why should be initiated 18.

Our comment: The reasonable explanation can be analytical error during glucose determination by enzymatic Sigma-Aldrich assay kit. However, this discrepancy was in the range of 10 % compared to the initial glucose concentration.

Figure 2. Heterotrophic cultivation of E. gracilis and paramylon production in stirred tank bioreactor by the different bioprocesses modes: a) fed batch (DW should start 22 g/L whereas other processes b, and c start at 32 g/L dry weight; b) repeated batch; c) continuous.

Our comment: The possible explanation can be the CSS heterogeneity as a medium constituent. Furthermore, it has to be pointed out that CSS and all other complex medium constituents were always homogenised before they were used for medium preparation. Bioreactor sterilization was done by indirect steam and therefore steam condensation effect in medium can be neglected. (see lines 263-267)

The discussion and conclusion should be strengthened.

Our comment: The discussion and conclusion are strengthened (see lines 169-171; 175-178; 207-232; 246-250; 264-269; 276-277; 318-322).

Reviewer 2 Report

The manuscript entitled “Heterotrophic cultivation of Euglena gracilis in stirred tank bioreactor: A promising bioprocess for sustainable paramylon production” reports the application of industrial byproduct corn steep solid to E. gracilis cultivation with various processes. It is informative to get insights of the cultivation process scale-up. However, some critical issues should be fixed prior to acceptance.

1. More references are required to bolster the author’s statements. There are many helpful studies that used complex medium (industrial byproducts) for heterotrophic microalgae cultivation.

2. Could you provide more specific information on reactors? For example, volume scale or size of the bioreactor in the Section 2.2. And the same inoculum volume was applied for both w.v. 5 L and w.v. 10 L cultivation? Since the assessment of the bioprocess conditions (batch, continuous, etc.) is the most important factor in the manuscript, more description on the relative information in the M&M section (instead of placing it in the Discussion section) would be necessary for the readers.

3. Was there any cell loss during continuous cultivation? Did you consider any evaporation losses during the cultivation experiments?

4. Authors should provide specific number of biomass and paramylon production results in the Results and Discussion section. For example, there are neither biomass (X) nor paramylon (P) concentration results described in the Section 3.1 so that clear comparison among the carbon sources cannot be seen. Additionally, vague expression on the results makes it impossible to get reasonable discussion comparing other literatures.

5. Discussion on why the complex medium (20 g/L glucose and 25 g/L CSS) was efficient would be required.

6. Confused expression

- Usage of the term ‘repeated batch’ and ‘repetitive batch’ would make the readers confusing.

7. Formatting

- Reference section should be revised according to the Journal’s guide and format for Authors. To be specific, first name and family name of the authors should be corrected. And the references on the list should be either numbered or not.

- Parentheses in Line #217 and #268-269 seem not necessary.

- Font sizes are required to be corrected between Line #267 and Line #278.

8. Typo

- Line #16: g/Lh à g/L/h

- Line #32: glucanes à glucan

- Line #208: E gracilis à E. gracilis

Author Response

Dear Reviewer,

we would like to give thanks to reviewer for the insightful comments and suggestions which were very useful during revision process. In general, the whole manuscript was carefully read and corrected (major changes are in red color) according to the reviewer’s suggestions. New references are added and consequently according to the authors instructions. Our comments to reviewer’s suggestions are presented below.

Reviewer 2:

The manuscript entitled “Heterotrophic cultivation of Euglena gracilis in stirred tank bioreactor: A promising bioprocess for sustainable paramylon production” reports the application of industrial byproduct corn steep solid to E. gracilis cultivation with various processes. It is informative to get insights of the cultivation process scale-up. However, some critical issues should be fixed prior to acceptance.

  1. More references are required to bolster the author’s statements. There are many helpful studies that used complex medium (industrial byproducts) for heterotrophic microalgae cultivation.

Our comment: Thank you for the suggestion. As per your advice we have incorporated more references (see lines 249-352; 356-357; 382-383; 393-395; 399-402).

  1. Could you provide more specific information on reactors? For example, volume scale or size of the bioreactor in the Section 2.2. And the same inoculum volume was applied for both w.v. 5 L and w.v. 10 L cultivation? Since the assessment of the bioprocess conditions (batch, continuous, etc.) is the most important factor in the manuscript, more description on the relative information in the M&M section (instead of placing it in the Discussion section) would be necessary for the readers.

Our comment: Thank you for the suggestion. We engaged more information of the cultivation conditions in the materials and methods section (see lines 94-99; 103-105;107-108; 111-112; 114-116; 118-120).

  1. Was there any cell loss during continuous cultivation? Did you consider any evaporation losses during the cultivation experiments?

Our comment: We agree with this comment. In this research the cell losses were on the negligible levels. During this investigation the water evaporation losses were also noticed and consequently every 24 hours in the STR was added approximately 250 mL of sterile water. (see lines 159-161)

  1. Authors should provide specific number of biomass and paramylon production results in the Results and Discussion section. For example, there are neither biomass (X) nor paramylon (P) concentration results described in the Section 3.1 so that clear comparison among the carbon sources cannot be seen. Additionally, vague expression on the results makes it impossible to get reasonable discussion comparing other literatures.

Our comment: Specific number of biomass and paramylon concentrations are given in section 3.1. (see lines 179, 80,189, 198, 206). Comparison among the carbon sources are given in the Table 1.

  1. Discussion on why the complex medium (20 g/L glucose and 25 g/L CSS) was efficient would be required.

Our comment: We agree with this comment and consequently discussion on the optimal complex medium composition with 20 g/L glucose and 25 g/L CSS is adapted from the previous investigations (see lines 207-224).

  1. Confused expression

- Usage of the term ‘repeated batch’ and ‘repetitive batch’ would make the readers confusing.

Our comment: We agree with this comment and consequently expression “repetitive batch” was replaced with expression “repeated batch” through the whole manuscript.

  1. Formatting

- Reference section should be revised according to the Journal’s guide and format for Authors. To be specific, first name and family name of the authors should be corrected. And the references on the list should be either numbered or not.

- Parentheses in Line #217 and #268-269 seem not necessary.

- Font sizes are required to be corrected between Line #267 and Line #278.

Our comment: We agree with these comments and consequently changes in the manuscript were made.

  1. Typo

- Line #16: g/Lh à g/L/h

- Line #32: glucanes à glucan

- Line #208: E gracilis à E. gracilis

Our comment: We agree with this comment and consequently changes in the manuscript were made.

We hope that current form of our manuscript is suitable for consideration to be published in journal Molecules.

Yours sincerely,

Tonči Rezić

Round 2

Reviewer 1 Report

The authors have made the desired corrections for the article.

Reviewer 2 Report

The manuscript is well-revised corresponding to the previous comments were well-revised. Just one thing necessary to be fixed is the reference citation in the text. Some citations are incorrectly written: for example, 'Jee Young et al., 2021' would certainly be 'Kim et al., 2021'; 'Min Seo et al., 2020' and 'Sunah et al., 2021' should be corrected as well. Additionally, 'Gissibil et al., 2017' in Line #224 needs to be corrected since it is not on the reference list.